# Systematic Booster after Rabies Pre-Exposure Prophylaxis to Alleviate Rabies Antibody Monitoring in Individuals at Risk of Occupational Exposure

**DOI:** 10.3390/vaccines9040309

**Published:** 2021-03-24

**Authors:** Perrine Parize, Jérémie Sommé, Laura Schaeffer, Florence Ribadeau-Dumas, Sheherazade Benabdelkader, Agnès Durand, Arnaud Tarantola, Johann Cailhol, Julia Goesch, Lauriane Kergoat, Anne-Sophie Le Guern, Marie-Laurence Mousel, Laurent Dacheux, Paul-Henri Consigny, Arnaud Fontanet, Beata Francuz, Hervé Bourhy

**Affiliations:** 1Institut Pasteur, Unit Lyssavirus Epidemiology and Neuropathology, National Reference Center for Rabies and WHO Collaborating Centre for Reference and Research on Rabies, 75015 Paris, France; florence.ribadeau-dumas@intradef.gouv.fr (F.R.-D.); sherazben@yahoo.fr (S.B.); arnaud.tarantola@santepubliquefrance.fr (A.T.); lauriane.kergoat@pasteur.fr (L.K.); laurent.dacheux@pasteur.fr (L.D.); herve.bourhy@pasteur.fr (H.B.); 2Institut Pasteur, Occupational Health Department, 75015 Paris, France; somme.j@chu-toulouse.fr (J.S.); marie-laurence.mousel@pasteur.fr (M.-L.M.); beata.francuz@pasteur.fr (B.F.); 3Institut Pasteur, Emerging Diseases Epidemiology Unit, Centre for Global Health Research and Education, 75015 Paris, France; laura.schaeffer@pasteur.fr (L.S.); arnaud.fontanet@pasteur.fr (A.F.); 4Laboratoire Cerballiance, 75017 Paris, France; agnes.durand@cerballiance.fr; 5Institut Pasteur, Centre Médical, Centre d’Infectiologie Necker-Pasteur, 75015 Paris, France; johann.cailhol@aphp.fr (J.C.); julia.goesch@pasteur.fr (J.G.); paul-henri.consigny@pasteur.fr (P.-H.C.); 6Institut Pasteur, Laboratory of the Medical Center, 75015 Paris, France; anne-sophie.le-guern@pasteur.fr; 7Conservatoire National des Arts et Métiers, 75003 Paris, France

**Keywords:** rabies, pre-exposure prophylaxis, humoral immunity, booster immunization, occupational health

## Abstract

Pre-exposure rabies prophylaxis (PrEP) is recommended for people at frequent or increased risk of professional exposure to lyssavirus (including rabies virus). PrEP provides protection against unrecognized exposure. After the primary vaccination, one’s immune response against rabies may decline over time. We aimed to evaluate the immune response to rabies in individuals immunized for occupational reasons before and after a booster dose of the rabies vaccine. With this aim, we retrospectively documented factors associated with an inadequate response in individuals vaccinated for occupational purposes. Our findings analyzed data from 498 vaccinated individuals and found that 17.2% of participants had an inadequate antibody titration documented after their primary vaccination without the booster, while inadequate response after an additional booster of the vaccine was evidenced in 0.5% of tested participants. This study showed that a single booster dose of vaccine after PrEP conferred a high and long-term immune response in nearly all individuals except for rare, low responders. A systematic rabies booster after primary vaccination may result in alleviating the monitoring strategy of post-PrEP antibody titers among exposed professionals.

## 1. Introduction

Rabies is a viral zoonosis responsible for approximately 59,000 human deaths each year, affecting mainly poor and rural populations of Asia and Africa [1]. The disease in humans is mainly transmitted by dogs through bites, scratches, contamination of the mucous membrane, or broken skin with saliva [2]. Rabies is 100% preventable after exposure to a rabid animal by the timely and adequate administration of postexposure prophylaxis (PEP) which consists of rabies vaccines and immunoglobulin in severe exposures [3]. In the absence of PEP, however, an infection can occur, and rabies acute encephalitis or meningoencephalitis can develop. The outcome of these complications is almost invariably fatal [4].

Pre-exposure rabies prophylaxis (PrEP) consists of a series of intramuscular or intradermal injections of rabies vaccine that primes the immune system and enables a prompt and robust anamnestic immune response after the booster doses [5]. PrEP may protect against rabies for individuals with unrecognized exposure and simplifies the postexposure regimen in case of documented exposure. Indeed, when PrEP has been administered even long before exposure, the PrEP regimen includes only two vaccination sessions three days apart. Moreover, immunoglobulins are unnecessary [6]. PrEP is recommended for people at frequent or increased risk for exposure to the rabies virus and other lyssaviruses. These individuals include laboratory workers dealing with lyssaviruses (individuals involved in rabies research, rabies diagnosis, or rabies biologics production), veterinarians and individuals working in contact with wildlife including bats, and, to a lesser degree, individuals working or traveling in high-risk areas [7].

The World Health Organization (WHO) recommendations regarding PrEP, the frequency of booster vaccinations, and the serological surveillance for at-risk individuals have changed since the first WHO expert consultation on Rabies report in 2005 [8]. Since 2013, a systematic booster is no longer recommended at one year after complete primary pre-exposure vaccination with the rabies vaccine [9]. After a complete primary vaccination, a neutralizing antibody titration is now required every six months, for up to two years, for individuals with potential occupational exposure. A booster dose of vaccine is indicated only if the antibody titer falls below the 0.5 IU/mL threshold which is considered a proxy for protection [6]. The recent literature, however, emphasizes the importance of a systematic booster after the primary vaccination to maintain a robust immune response against rabies [5,10,11,12]. Moreover, biannual antibody titration in laboratory workers is challenging due to low compliance and poor access to titration in low-income countries. Periodic booster injections every 1–2 years are recommended if serological testing is unavailable; however, this is rarely achievable.

This study aimed to evaluate the immune response to rabies PrEP in individuals immunized for occupational purposes before and after a booster dose of the rabies vaccine. It also sought to document factors associated with an inadequate response before receiving the booster.

## 2. Materials and Methods

### 2.1. Ethical Statement

The study protocol was approved by the French Ethical Committee (Comité de Protection des Personnes Ile de France II n° 2001-532 RCRB). According to the EU regulations on the protection of personal data, all individuals received written information on the study and use of their personal data for research purposes. The data were extracted retrospectively by reviewing both electronic medical records and paper charts. All data and results were anonymized.

### 2.2. Population and Data

The study population consisted of laboratory workers of the Institut Pasteur of Paris, vaccinated against rabies by the Occupational Health Service because they were working in contact with lyssaviruses as well as Non-Governmental Organization (NGO) workers who were vaccinated at the Institut Pasteur vaccination center before their missions in rabies-enzootic countries. All laboratory or NGO workers were included if: (1) They had received a complete rabies PrEP (three sessions of one intramuscular dose each, using a Vero cell rabies vaccine or a purified chick embryo cell vaccine); or (2) They had at least one post-PrEP antibody titration. Inclusion criteria for this study limited the sample to those who had at least one post-PrEP antibody titration before and/or after a first vaccine booster dose between 1 January 2000–1 October 2015 for laboratory workers and between 1 January 2007–1 October 2015 for NGO workers. Individuals aged less than 18 years old, those reporting a former rabies postexposure prophylaxis, or those who had received more than three intramuscular doses of rabies vaccine were excluded from the study.

The following information was extracted from the participants’ medical records: Occupation (laboratory or NGO worker); year of birth; sex at birth; number of doses of vaccine received and dates of vaccination; date of rabies antibody titration and test results; other vaccines administered concurrently with rabies vaccine; specific medical conditions; and treatment by chloroquine or infectious disease episode concurrent to rabies primary vaccination.

### 2.3. Postvaccination Rabies Antibodies Titration

Postvaccination rabies antibodies titration was carried out on an outpatient basis in various medical laboratories using the PLATELIA™ RABIES II ELISA method (Bio-Rad Laboratories, Hercules, CA, USA) [13], an enzyme-linked immunosorbent assay based on the detection and titration of the anti-glycoprotein antibodies. The principle of the technique described by the manufacturer is as follows: Purified G glycoprotein was diluted in sodium carbonate buffer (0.1 M, pH 9.5) and used for coating of 96-well microdilution plates (overnight incubation at room temperature). Residual adsorption sites on the plates were saturated at room temperature by incubating them with a phosphate buffer (0.1 M, pH 7.4) which was supplemented with skim milk powder. The ELISA test was performed as described in the package insert supplied by Bio-Rad. Reagents were stored at 2–8 °C and placed at room temperature for at least 30 min before use. Briefly, the sample was prediluted 1:100 in sample buffer, and 100 μL was incubated in a microplate well sensitized with the rabies virus glycoprotein for 60 ± 5 min at 37 ± 1 °C. One negative (R3) and two positive controls (R4a and b) were tested in each run. The negative control was made of synthetic material, and the positive controls were made of therapeutic rabies immunoglobulins in synthetic material. The positive controls were calibrated against the WHO international standard for rabies immunoglobulin. R4b (4 equivalent units (EU)/mL) was used to establish a reference curve after successive two-fold serial dilutions (S5 = 2 EU/mL, S4 = 1 EU/mL, S3 = 0.5 EU/mL, S2 = 0.25 EU/mL, S1 = 0.125 EU/mL). The test can be used as a quantitative (with the use of R3, R4a, R4b controls and preparation of the serial dilutions S1–S5 from R4b) method. After three washing cycles with 1× washing solution with a microplate washer, horseradish peroxidase-conjugated protein A was incubated 1 h ± 5 min at 37 ± 2 °C in a microplate incubator. Following five washes, the linked peroxidase conjugate was visualized with 3,3′,5,5′-tetramethylbenzidine (TMB) incubated for 30 ± 5 min at room temperature. The enzyme reaction is stopped by addition of 1N sulfuric acid solution. Absorbance was measured at 450–620 nm with the use of a microplate reader with the specific rabies program. The dose–optical density response curve can be used to determine the titer of each serum. Sera titers were expressed in equivalent unit per mL (EU/mL). According to the WHO, an “adequate” rabies antibody titer likely confers protective immunity against rabies virus infection, and is defined as ≥0.5 IU/mL [6]. By analogy, we considered a titer of 0.5 EU/mL as the threshold determined by ELISA [6]. Further, the maximum rabies antibody level measured by the ELISA method was 4 Equivalent (E)U/mL; levels higher than this value were reported as >4 EU/mL.

### 2.4. Statistical Analysis

Baseline and demographic characteristics were summarized using median (interquartile range) and percentages for continuous and categorical variables, respectively. Characteristics were compared to identify predictive factors of inadequate rabies antibody titers <0.5 EU/mL. Univariate analyses were performed using Student *t*-test or Mann-Whitney test for continuous variables, and the Pearson χ^2^ test or Fisher exact test were utilized for categorical variables, as appropriate. All factors significantly associated with the outcome at *p* < 0.25 in the univariate analysis were included in the multivariate model. Multivariate analysis was performed using logistic regression. Confidence intervals at the 95% level were reported for each adjusted odds-ratio (OR). All tests were two-tailed, and statistical significance was set at <0.05. Data were analyzed using STATA IC version 13 (StataCorp LP, College Station, TX, USA).

## 3. Results

### 3.1. Baseline Characteristics

During the study period, 509 individuals met the inclusion criteria. We were unable to provide written information to 11 individuals of the 498 that were included (Figure 1). A total of 738 blood test results were available for assessment. The baseline characteristics of the 498 individuals included in the study are shown in Table 1. Among participants, 150 (30.1%) were laboratory workers and 348 (69.9%) were nongovernmental organization (NGO) workers (mainly healthcare workers) sent on missions in rabies-enzootic countries. Women represented 259/498 (52%) of the population, and the median age at primary vaccination was 32 years (IQR 27–39 years). A specific medical condition was documented in six individuals: Diabetes mellitus (*n* = 2); selective immunoglobulin A deficiency (*n* = 1); untreated HIV infection (*n* = 1); chronic hepatitis C (*n* = 1); and treatment by tumor necrosis factor inhibitors (*n* = 1). No participant was treated by chloroquine, and no one reported an intercurrent infection during the primary rabies vaccination course.

### 3.2. Postprimary Rabies Vaccination Rresponse

A total of 355/498 (71.3%) individuals underwent antibody titration after PrEP and before boosting, including 60 participants who had two or more blood tests postprimary PrEP (with a maximum of six blood tests per participant). Between dates ranging from D_26_ to D_4596_ following PrEP (with D_0_ being the day of the first dose of the schedule), 452 blood tests were performed. An adequate response on all periodic tests was evidenced for 294 (82.8%) participants (377 tests, including 102 which assessed for a titer above 4 EU/mL), while an inadequate titer was documented at least once for 61 (17.2%) individuals (Figure 2). Among the latter, 52 (85.2%) had an inadequate titer on the first post-PrEP serology, whereas declining titers were observed in nine individuals (first titer adequate subsequently waning below 0.5 EU/mL, as revealed by periodic monitoring). There were 438 days between the median delay between PrEP and the first test showing inadequate titers (IQR 326–1215 days). Only three individuals (four tests) had inadequate titers documented in the first six months following PrEP, representing 2.1% of the 141 patients tested and 2.8% of the 143 tests performed during the study period. However, the number of participants with inadequate titers rose to 19/66 (28.8%) and 15/65 (23.1%) between 6–12 and 12–18 months following PrEP, respectively.

Multivariate analyses assessed factors that were significantly and independently associated with an inadequate titer on the first post-PrEP serology and before vaccine booster dose. These factors included male sex at birth (OR 3.85; 95% CI 1.86–7.93), an interval between primary vaccination and serology of >6 months (OR 9.5; 95% CI 2.70–33.36 between 6 and 24 months and 8.47; 95% CI 2.37–30.30 after 24 months), and the simultaneous administration of a nonrabies vaccine during PrEP (OR 2.36; 95% CI 1.11–5.01) (Table 2). Age at PrEP and occupation (laboratory or NGO worker) were not significantly associated with inadequate response.

### 3.3. Postbooster Rabies Vaccine Response

Overall, 220/498 (44.2%) individuals underwent antibody titration after a first rabies booster dose, including 35 patients who had more than one test after the booster (maximum of six tests per individual). The median delay between the initial PrEP and the first booster was 490 days (min 93; max 4181). A total of 286 postbooster antibody titers were measured in 220 participants with the delay between the first booster and titration ranging from 3 to 6281 days.

A nonprotective titer was found for only one participant after the booster (Figure 3). This 27-year-old man had an inadequate titer at D_124_ post-PrEP, and received a booster dose one year post-PrEP with a protective antibody titer at D_30_. Antibody waning below 0.5 EU/mL was evidenced one year after the booster dose. Apart from this participant, all of the other 219 individuals exhibited adequate postbooster titers, including on tests performed more than 5 or 10 years after a single booster. Among these individuals with adequate titers, 155 (70.8%) were assessed as > 4 EU/mL on at least one test.

Six patients with specific medical conditions presented an adequate rabies antibody titer. Titration was performed post-PrEP for the participant with immunoglobulin A deficiency postprimary and postbooster for the remaining sample.

## 4. Discussion

We reported our experience related to rabies pre-exposure prophylaxis and antibody titer monitoring among 498 individuals vaccinated for occupational purposes. Our findings indicated that 28.8% of tested individuals exhibited inadequate postvaccination titers when assessed between 6 and 12 months following PrEP. Male sex at birth, a time-lapse greater than six months between PrEP and antibody titration, and simultaneous nonrabies vaccination were independently associated with titers considered nonprotective on the first post-PrEP assessment. Furthermore, our results show that a single booster dose of vaccine after PrEP confers a high and long-term immune response in nearly all individuals with the exception of rare low responders.

Several other studies have demonstrated the decline of the rabies virus neutralizing antibodies titer following rabies PrEP [11]. Dougas et al. reported inadequate rabies antibody titers in 26.4% of 144 high-risk professionals tested >90 days after PrEP [12], Banga et al. in 29.4% of 603 US veterinary students two years after PrEP [14] and Lim et al. in 39.4% of 66 individuals vaccinated for occupational purposes in Singapore one year after PrEP [10]. In our study, rates of documented inadequate responses increased significantly in the six months after PrEP, ranging from 23.1% to 28.8% of the participants tested between 6 and 18 months following PrEP. This percentage of inadequate responders was substantial among professionals who were frequently exposed to rabies at work.

Inadequate antibody titers were independently associated with male sex at birth in the present study. This result is consistent with the findings of several other studies that reported higher antibody titers after rabies PrEP or postexposure prophylaxis in females [11,12,14,15]. A stronger postvaccination antibody response in females has been evidenced with other vaccines such as hepatitis A, hepatitis B, rubella, diphtheria, tetanus, and brucella [16], whereas the response was stronger in males for other vaccines such as pneumococcal, meningococcal vaccines, measles, and yellow fever. Sex differences in humoral response have been poorly evaluated in vaccine trials and, thus, explanations remain unclear [17]. Differences may not be completely explained by gonadal hormones that affect the immune response in females [18]. A higher CD4 + T cell count and/or a stronger Th2 response in females may also play a role in these differences. It is unclear whether the difference in rabies antibody titers between females and males is of any clinical significance.

Several authors have shown that both higher seroconversion rates and higher antibody titers were observed in younger age groups [15,19,20]. However, no statistical association between age at PrEP and immune response was evidenced in our study. It should be noted that the majority of the study population were young adults (median age 32, IQR 27–39 years), preventing us from detecting age-related associations.

To our knowledge, our study is the first suggesting that the simultaneous administration of a nonrabies vaccine during PrEP could be associated with an inadequate immune response against rabies. The simultaneous administration of vaccines is widely promoted because it increases the probability that an individual receives the full range of vaccines appropriate for their age or in the case of occupational or pretravel vaccinations. Most inactivated and live, attenuated vaccines can be administered simultaneously at distinct anatomical sites without impairing antibody responses or increasing rates of adverse reactions. There are some exceptions, specifically, the quadrivalent meningococcal conjugate vaccine + pneumococcal conjugate vaccine, as well as the pneumococcal conjugate vaccine + pneumococcal polysaccharide vaccine [21]. The concurrent administration of these vaccines is not recommended, as a stronger immune response is expected when they are administrated separately. Furthermore, it has been shown that the immune response to the second vaccine may be impaired when two live, attenuated vaccines are administered less than 28 days apart [21]. Our study failed to identify the types of vaccines specifically associated with the decreased response to rabies vaccination due to the diversity of vaccines administrated simultaneously with rabies PrEP in participants and the limited size of the study. The possible impairment of post-PrEP response by concurrent administration of other vaccines needs to be explored further in other studies designed specifically to address this point.

Our study demonstrates that a booster dose of rabies vaccine confers a robust and long-term immune response. This finding is consistent with the results of numerous other studies, emphasizing the importance of a systematic booster after rabies PrEP to maintain antibody titers considered protective [5,10,11,12]. In our study, only one participant out of 220 monitored postbooster demonstrated a titer considered nonprotective one year after a booster dose of the vaccine. Low responders demonstrate a low postvaccination titer and a rapid antibody decline. Individuals with occupational, continuous, or frequent risk of exposure to lyssaviruses in settings where serological monitoring is available could benefit from the titration of rabies antibody six months after PrEP. This screening would be able to detect the few low responders. This group of low responders would benefit from a tailored monitoring strategy with serological monitoring every 1–2 years and booster doses whenever antibody titers fall below 0.5 EU/mL. For all other individuals presenting an adequate six-month titer, a systematic booster between six months and one year post-PrEP, a subsequent serological monitoring every two years to address continuing exposure risk, and a titration every five years in case of frequent, but intermittent, exposure risk may suffice. In contrast, a rabies vaccine booster dose administered 6 to 12 months after PrEP followed by once every five years could ensure adequate antibody titers in over 99% of people in settings where serological testing is not readily available. We believe these strategies would be more acceptable and realistic for workers, especially those in settings where serological monitoring is not routinely available and where periodic booster injections every 1–2 years are rarely achievable due to the cost of the vaccine.

Our study has several limitations. First, the humoral response is only one aspect of the immune response to the rabies vaccination: cell-mediated immune responses of the rabies virus are not yet clearly understood, and thus, could not be easily assessed [22]. We considered the measurement of rabies antibodies as the best available method to assess postvaccination immune response, a proxy for protection, as per recommendations. Second, we used an ELISA method to determine the postvaccination rabies antibody titers, although the WHO recommends the use of the rapid fluorescent focus inhibition test (RFFIT) or fluorescent antibody virus neutralization (FAVN) [6]. In France, RFFIT is only available at the National Reference Laboratory for Rabies, and is not used routinely. Serology results using the ELISA method, however, are comparable to those of the RFFIT method. Moreover, ELISA is considered to be a reliable alternative when RFFIT is unavailable [6,13,15,23]. Several studies on antibody response to rabies vaccination in individuals with potential occupational exposure have been performed using an ELISA test [10,12], and most people vaccinated for occupational purposes worldwide have rabies antibody titration monitoring assessed using this method. Third, we observed limitations inherent to observational studies. Records were not designed for the study, and some data were unavailable. This limitation, however, resulted in few missing data regarding PrEP (type of vaccine, vaccination schedule) or follow-up of individuals (date of rabies antibody titrations and boosters) and reflects real-world, clinical follow-up of workers exposed to lyssaviruses. Fourth, response to rabies vaccination was not assessed in children or the elderly in our study because the study participants were people of working age vaccinated for occupational purposes. The absence of these two age groups may have prevented us from detecting an association between age and post-PrEP antibody response. Lastly, the PrEP schedule assessed in this study comprised three intramuscular rabies vaccine injections. However, an abridged PrEP regimen, consisting of only two visits (1-site IM regimen or 2-site ID regimen on days 0 and 7), was recommended in the most recent WHO position paper in 2018, as this regimen produces similar, consistent antibody responses as the classical regimen (3 visits—day 0, 7, and 21 or 28). This new schedule could not be assessed in our study, as we included only individuals immunized before 2015.

The strength of our study was the assessment of post-PrEP rabies antibody titers in a population representative of individuals vaccinated for occupational purposes in real-world conditions. Participants with varying occupational profiles were included as well as individuals with different vaccine regimens, including concurrent nonrabies vaccine administration. Finally, our study population included a large proportion of female participants, a group that is too often underrepresented in vaccine studies.

Periodic serological assessments and rabies boosters are not needed in travelers after a rabies PrEP, as unrecognized exposures are rare in this population, and postexposure prophylaxis with two boosters D_0_–D_3_ results in adequate response, even decades after PrEP. People at continual, frequent, or increased risk for exposure to the rabies virus and others lyssaviruses due to their occupation, however, need to achieve adequate antibody titers permanently. Our study shows the importance of a systematic booster between six months and one year following PrEP in these individuals. This booster dose would ensure the maintenance of a robust immune response against rabies, protecting workers against unrecognized exposures, and would lead to the alleviation of the monitoring strategy of post-PrEP antibody titers. Our results have practical applications regarding recommendations for occupational PrEP administration and antibody monitoring.

## Figures and Tables

**Figure 1 vaccines-09-00309-f001:**
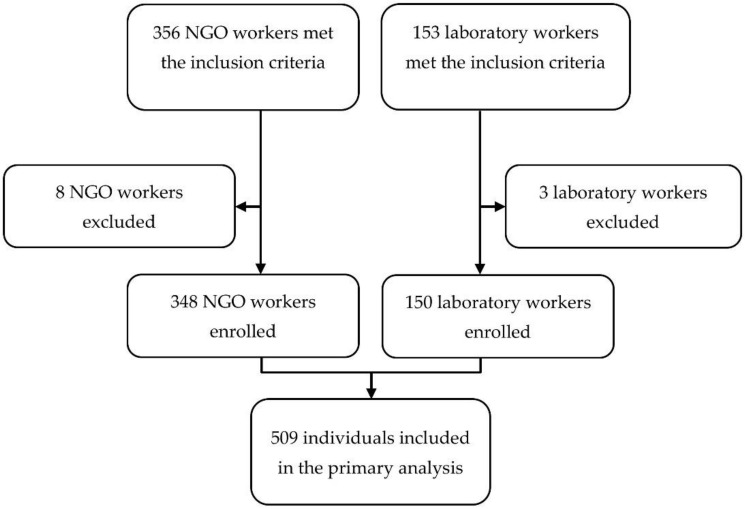
Study flow chart.

**Figure 2 vaccines-09-00309-f002:**
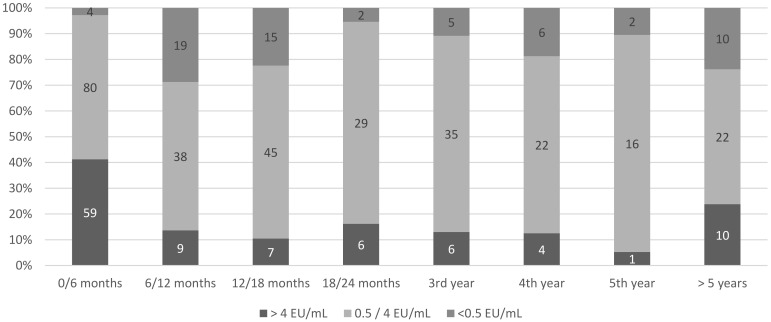
Percentage and number of antibody titers assessed after rabies primary vaccination by titer categories and time-lapse since primary vaccination, in individuals vaccinated for occupational purposes (*n* = 452), Institut Pasteur, Paris 2000–2015.

**Figure 3 vaccines-09-00309-f003:**
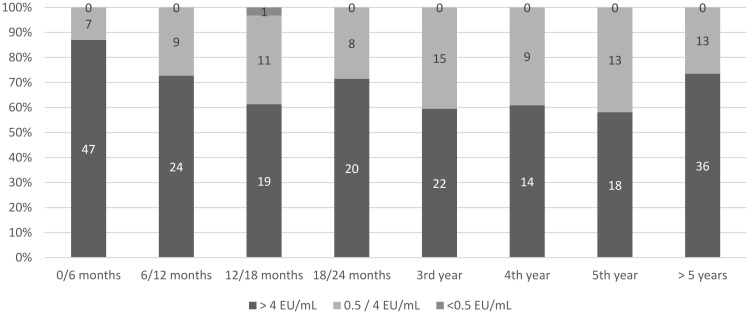
Percentage and number of antibody titers assessed after rabies primary vaccination and a single booster by titer categories and time-lapse since booster, in individuals vaccinated for occupational purposes (*n* = 286), Institut Pasteur, Paris 2000–2015.

**Table 1 vaccines-09-00309-t001:** Characteristics of the study population (*n* = 498), Institut Pasteur, Paris 2000–2015.

Variable	*n* (%)	Median (Interquartile Range)
Female sex at birth	259 (52%)	
Age (years)		32 (27–39)
Laboratory workers	150 (30.1%)	
NGO workers	348 (69.9%)	
Specific medical conditions	6 (1.2%)	
**Pre-exposure rabies Prophylaxis (PrEP)**	**498 (100%)**	
Interval between first and last dose of vaccine		23 (21–28) *
Simultaneous administration of a nonrabies vaccine	234 (47%)	
Concurrent treatment by chloroquine	0	
**Individuals with post-PrEP serology**	**355 (71.3%)**	
Number of tests	452	
Time between PrEP and first titration (days)		351 (72–865)
Individuals with inadequate first titers	52 (14.6%)	
Individuals with at least one inadequate titer during follow-up	61 (17.2%)	
Time between primary vaccination and first inadequate titer (days)		438 (326–1215)
**Individuals with postbooster serology**	**220 (44.2%)**	
Number of tests	286	
Time between PrEP and first booster (days)		490 (398–842)
Time between booster and first postbooster titer (days)		636 (195–1517)
Individuals with at least one inadequate titer during follow-up	1 (0.5%)	

* Missing data for 31 individuals; Inadequate titers defined as below the 0.5 IU/mL; titers considered a proxy for protection.

**Table 2 vaccines-09-00309-t002:** Variables associated with antibody titers considered protective (“adequate”) or nonprotective (“inadequate”) on first assessment after rabies pre-exposure prophylaxis (univariate and multivariate analyses), Institut Pasteur, Paris 2000–2015.

Variables	Inadequate Titer <0.5 IU/mL (*n* = 52)	Adequate Titer ≥0.5 IU/mL (*n* = 303)	*p*-Value	Adjusted Odds Ratio (95% Confidence Interval)	*p*-Value
**Sex at birth**			<0.001		<0.001
Female	12 (23.1)	175 (57.8)		1 (ref)	
Male	40 (76.9)	128 (42.2)		3.85 (1.86–7.93)	
**Age at PrEP (years)**			0.10		0.29
<27	7 (13.5)	86 (28.4)		1 (ref)	
27–32	14 (26.9)	84 (27.7)		1.81 (0.65–5.06)	
33–38	15 (28.8)	61 (20.1)		2.63 (0.93–7.40)	
>38	16 (30.8)	72 (23.8)		2.27 (0.81–6.35)	
**Occupation**			<0.001		0.67
Laboratory workers	6 (11.5)	132 (43.6)		1 (ref)	
NGO workers	46 (88.5)	171 (56.4)		1.30 (0.39–4.35)	
**Duration between first and last PrEP dose**			0.13		
<21 days	4 (7.7)	15 (5.0)			
22–28 days	39 (75.0)	209 (69.0)			
>28 days	9 (17.3)	56 (18.5)			
**Time between first PrEP dose and serology**					
<6 months	3 (5.8)	133 (43.8)	<0.001	1 (ref)	0.002
6–24 months	28 (53.8)	85 (28.1)		9.50 (2.70–33.36)	
>24 months	21 (40.4)	85 (28.1)		8.47 (2.37–30.30)	
**Nonrabies vaccine concurrent with rabies PrEP**			0.002		0.03
Yes	35 (67.3)	133 (43.9)		1 (ref)	
No	17 (32.7)	170 (56.1)		2.36 (1.11–5.01)	

## Data Availability

The data presented in this study are available on request from the corresponding author.

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
