# Peer review of "Systematic Booster after Rabies Pre-Exposure Prophylaxis to Alleviate Rabies Antibody Monitoring in Individuals at Risk of Occupational Exposure"

_vaccines, 2021, doi:10.3390/vaccines9040309_

Round 1
Reviewer 1 Report
The authors investigate the longevity of Rabies Pre-exposure prophylaxis vaccine in humans. The study found that approximately a quarter of the vaccinees did not have adequate antibody responses 6-18 months post-vaccination. This number was significantly reduced by providing a booster vaccine. The study is informative and the data are well presented.
Two minor comments:
1) Consider adding a section in the "Material and Methods" on the antibody titer procedure.
2) "Long-lasting" is an ill-defined term and is not appropriate. Please consider rewording throughout the manuscript.
Author Response
Reviewer 1
- Consider adding a section in the "Material and Methods" on the antibody titer procedure.
I would like to thank the reviewer for this useful comment. The principle of the technique described by the manufacturer has been added to the section “Material and Methods”.
2) "Long-lasting" is an ill-defined term and is not appropriate. Please consider rewording throughout the manuscript.
Thank you for this suggestion. The manuscript has been modified using the phrase “long-term” instead of “long-lasting”.
Reviewer 2 Report
This manuscript deals with the systematic booster after rabies pre-exposure prophylaxis (PrEP) to alleviate rabies antibody monitoring in individuals at risk of occupational exposure. They monitored the antibody titers among 498 individuals vaccinated for occupational purposes. Only three individuals had inadequate (<0.5EU/mL) titers in the first 6 months following PrEP. However, the number of participants with inadequate titers rose up to 28.8% in 6-12 months. Male sex at birth, a time lapse greater than six months between PrEP and antibody titration and simultaneous non-rabies vaccination were independently associated with titers considered non-protective on the first post-PrEP assessment.
They also showed that a single booster dose of vaccine after PrEP confers high and long-lasting immune response in nearly all individuals. This manuscript provided a practical strategies of booster vaccination in individuals immunized for occupational purpose.
The paper is well-written, and the authors clearly discussed the results obtained in this study. I think the paper is acceptable for publication after minor revision.
Figure 1. The arrow is out of alignment. Please adjust the lines.
Author Response
Reviewer 2
Figure 1. The arrow is out of alignment. Please adjust the lines.
Thank you for this comment, the figure has been adjusted following the reviewer recommendations.
Reviewer 3 Report
My minor grammatical comments are in the attached Word document. My only topical comment is that anecdotally I have heard that the titres of many carers vaccinated against rabies in Australia because they care for flying foxes actually rises over the years, similarly to the last cohort (> 5 years) and I have wondered whether this could be due to repeated low level exposure to the Australian Bat Lyssavirus in the bats they care for.

Author Response
Reviewer 3
My minor grammatical comments are in the attached Word document.
I would like to thank the reviewer for the suggestion of grammatical improvement. The text has been edited.
My only topical comment is that anecdotally I have heard that the titres of many carers vaccinated against rabies in Australia because they care for flying foxes actually rises over the years, similarly to the last cohort (> 5 years) and I have wondered whether this could be due to repeated low level exposure to the Australian Bat Lyssavirus in the bats they care for.
Thank you for sharing this interesting comment. We did not observe spontaneous rises of antibodies titres over years in our study population because participants at frequent risk of exposure were monitored closely and got the booster dose of vaccine as soon as their titres fell below 0.5 EU/ml.